# Global warming leads to larger bats with a faster life history pace in the long-lived Bechstein's bat (*Myotis bechsteinii*)

Carolin Mundinger [1] ✉, Toni Fleischer [2], Alexander Scheuerlein [1,3] & Gerald Kerth [1,3]

Whether species can cope with environmental change depends considerably on their life history. Bats have long lifespans and low reproductive rates which make them vulnerable to environmental changes. Global warming causes Bechstein's bats (*Myotis bechsteinii*) to produce larger females that face a higher mortality risk. Here, we test whether these larger females are able to offset their elevated mortality risk by adopting a faster life history. We analysed an individual-based 25-year dataset from 331 RFID-tagged wild bats and combine genetic pedigrees with data on survival, reproduction and body size. We find that size-dependent fecundity and age at first reproduction drive the observed increase in mortality. Because larger females have an earlier onset of reproduction and shorter generation times, lifetime reproductive success remains remarkably stable across individuals with different body sizes. Our study demonstrates a rapid shift to a faster pace of life in a mammal with a slow life history.

[1] Applied Zoology and Nature Conservation, Zoological Institute and Museum, University of Greifswald, Loitzer Straße 26, 17489 Greifswald, Germany. [2] Leipzig University Medical Center, Department of Psychiatry and Psychotherapy, Semmelweisstraße 10, 04103 Leipzig, Germany. [3]These authors contributed equally: Alexander Scheuerlein, Gerald Kerth. ✉email: c.mundinger@gmx.de

Climate change is predicted to cause global declines in animal populations, paving the way for high extinction rates as climate change advances[1]. At the same time, animals adapt to climate change with changes in morphology[2–4], behaviour[5], and phenology[6–8]. Some of these responses result from phenotypic plasticity, while others evolved through genetic adaptation[9]. Species with a slow life history are hypothesized to be particularly vulnerable to environmental change, as they cannot quickly respond by genetic adaptation[10,11]. Moreover, their populations take a long time to recover from crashes[1]. So far, it is largely unknown to which degree animals with a slow life history can cope with climate change by speeding up their 'pace of life'.

Life history theory aims to understand how the environment shapes resource allocation among competing vital functions, e.g., reproduction, growth, and body maintenance[12–15]. Life histories can be classified using pace of life measures, such as generation time, lifespan, or fecundity rate[16–18]. Although changes in life histories have been frequently documented, they are typically attributed to specific factors, such as predation pressure[19] or specific weather patterns[20], and focus on a single trait[21]. Long-term field studies on the effect of climate change on multiple life-history traits remain extremely rare, particularly in long-lived mammals[22]. To be able to predict the population persistence of species with a slow life history, it is crucial to understand whether changes in the pace of life are limited to evolutionary time scales that are extended in species with long generation times, or result from fast plastic responses[23].

Bechstein's bats (*Myotis bechsteinii*) can reach 21 years of age[24], and grow to larger adult body sizes in summers with higher temperatures during a sensitive period[25]. At the same time, larger female bats face an increased mortality risk[25,26]. This raises concerns about whether populations will persist if summers continue to get warmer as a result of global warming. If adult female mortality increases with climate change as predicted[27], the longevity and low reproductive rates of Bechstein's bats place them at high risk of extinction, unless larger females balance their higher mortality with a faster rate of reproduction.

Here, we assess how temperature-induced changes in size affect demographic rates in wild Bechstein's bats. Our aim is to investigate whether females of this long-lived species are able to respond to the consequences of global warming by adopting a faster life history. We use a longitudinal, individual-based 25-year dataset of 331 RFID-tagged bats, combining genetic pedigrees with long-term data on survival, reproduction and body size. We first assess whether size influences the reproductive rate and, specifically, age at first reproduction. We then evaluate the costs of early reproduction on future survival, and investigate the relationship between size and the pace of life. Finally, we compare individual lifetime reproductive success as a measure of fitness across individuals with slow and fast life histories. Because individual fitness is strongly associated with longevity in long-lived mammals[28–30], we expect to find a reduced fitness in larger females due to increased mortality[25,26].

Overall, our study reveals a rapid switch to a faster pace of life in a slow-reproducing, long-lived mammal species. Using long-term data, we show that body size positively influences annual fecundity, and especially age at first reproduction, which in turn is associated with long-term costs such as a higher mortality rate and reduced lifespan. Consequently, the effect of body size on mortality rate may be explained by the costs of an earlier onset of reproduction. Moreover, we show that an earlier start of reproduction together with higher reproductive rates during later life stages in larger bats, largely compensate for the negative effect of reduced longevity. As Bechstein's bats' offspring grow larger in warmer summers, global warming will likely further lead to bats

with larger body size, which will also accelerate the life history pace of these populations. This strategy may however only allow Bechstein's bats to cope with climate change if they live in high-quality habitats that allow for frequent breeding. Presumably, switching to a faster life history in response to global warming is a risky strategy when extreme weather events become more frequent, as is predicted by most climate change scenarios[31].

## Results

**Body size, age at first reproduction and fecundity.** Average forearm length of all adult females with complete life histories ($n = 331$) was 42.7 mm ($+/- 1.2$ mm SD). Adult body size is known to be affected by the summer minimum temperature during a critical growth period in June-July (see Mundinger et al. 2021[25]). Bats grow to larger sizes in warmer summers[25]. This relationship is robust, and can also be found in a subset of the data used in this study, where only females with complete life histories are included (see Fig. 1).

Among reproducing females ($n = 225$), the first reproduction event occurred at an average age of 2.2 years ($+/- 1.1$ years SD). Of all adult females, 32% never reproduced (those females died at an age of 1.7 years ($+/- 1.1$ years SD), with a mean forearm size of 42.8 mm).

Age at first reproduction was predicted by body size ($P = 0.003$; model 4, see Supplementary Table S1), with larger bats starting to reproduce at younger ages than smaller bats (see Fig. 2a). Age at first reproduction was also density-dependent ($P = 0.01$), with members of larger colonies starting to reproduce earlier (see Fig. 2b). Fecundity (= lifetime reproductive success (LRS) divided by the number of breeding seasons) was best explained by body size ($P = 0.003$; Fig. 2c) and age at first reproduction (P < 0.002; deviance explained=41.9%, Fig. 2d, and Supplementary Table S2), with higher fecundity rates occurring in larger bats and those that started reproduction at younger ages.

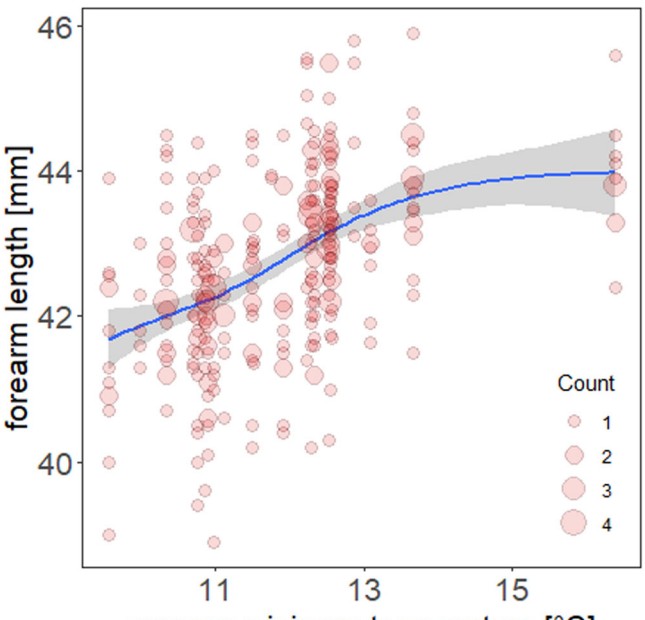

**Fig. 1 The influence of the summer minimum temperature during the birth year on the body size of bats.** The influence of the summer minimum temperature [°C] during a critical time window in the birth year on the body size of bats (all bats with complete life histories, $n = 331$). Count indicates the number of overlapping data points. The line and confidence interval indicate the smooth from a GAM of forearm length on summer temperature with colony ID and year as random factors.

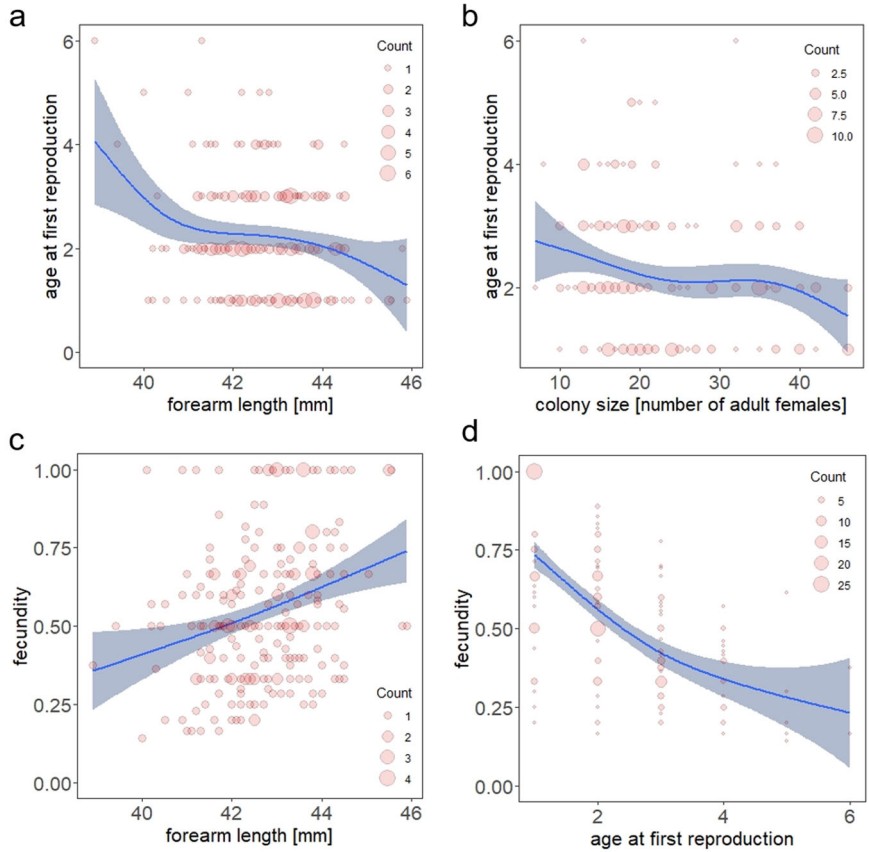

**Fig. 2 Impact of body size and colony size on the life history traits of age at first reproduction and fecundity.** Impact of (**a**) body size and (**b**) colony size (as the number of adult females in a colony) on the age of first reproduction (as age in years); and the impact of (**c**) body size and (**d**) age at first reproduction on the fecundity (as the lifetime reproductive success divided by the number of breeding seasons) as smooth functions of a GAM ($k = 6$, $n = 225$). Ninety-five per cent confidence intervals are added. Count gives the number of overlapping data points.

Females that bred in their first year of life died earlier (at an age of 3.9 years with a standard deviation of $+/- 3.4$ years) than late breeders (first reproduction at age 2: $5.5 +/- 2.9$ years; the combined group of first reproduction at age 3 and older: $7.2 +/- 3.2$ years; Kruskal-Wallis $\chi^2 = 44.88$, df $= 2$, $P < 0.001$; see Supplementary Fig. S1).

**Effects on mortality**. In the discrete survival analysis of long-term effects on mortality, the best model (14) explained 48.8 % of the deviance (see Supplementary Tables S3a and b). Age at first reproduction ($P < 0.001$), age ($P < 0.001$), and fecundity ($P < 0.001$) had significant effects on mortality, with older bats and bats with a higher fecundity showing a higher risk of mortality. Bats reproducing earlier had higher mortality risks than bats that started reproduction at a later age. Notably, body size did no longer influence the mortality hazard, after we included reproduction parameters.

**Determinants of fitness**. Mean LRS across all female adult bats with completed life histories ($n = 331$) was 1.98 ($+/- 2.21$) offspring. LRS was mostly determined by maximum age ($P < 0.001$) and age at first reproduction ($P < 0.001$) in the best model, that explained 78.9% of variance in LRS (see model 8, Supplementary Table S4). LRS was a nearly linear function of maximum age (Fig. 3a). Although body size was still included in this best model, it had no significant influence ($P = 0.09$), implying that overall LRS was similar across small and large bats (see Fig. 3b).

**Trajectories of mortality and reproduction**. Body size determined the rate of successful reproduction during the whole life of a bat. Larger individuals were characterized by a higher annual rate of reproduction throughout all ages (Fig. 4). Corroborating previous results[25], the general mortality risk increased with body size and age. When examining reproductive rate in a given year, the best model (model number 5; see Supplementary Table S5) found a significant positive impact of body size ($P < 0.001$) and age ($P < 0.001$). Colony size was not included in the best model, indicating that the reproductive rate was not density dependent.

**Body size and the pace of life**. Generation time varied across body sizes, with medium-sized bats (around 42 mm forearm length) having the longest generation time of about 5 years, and smaller bats having relatively similar generation time (Fig. 5a). In contrast, larger bats have much shorter generation times (about 4 years). As a consequence, only bats with a forearm size larger than 41.4 mm but smaller than 44.3 mm promote population growth ($\lambda > 1$, Fig. 5b).

## Discussion

In rapidly changing environments, species with slow life histories are considered to be particularly vulnerable to extinction[1]. Our long-term field study suggests that a long-lived mammal species may be able to cope with global warming by speeding up their pace of life. In our model species, warmer temperatures lead to larger bats with shorter lifespans, but these large bats have a higher chance of early reproduction and shorter generation times. Collectively, these changes allow lifetime reproductive success

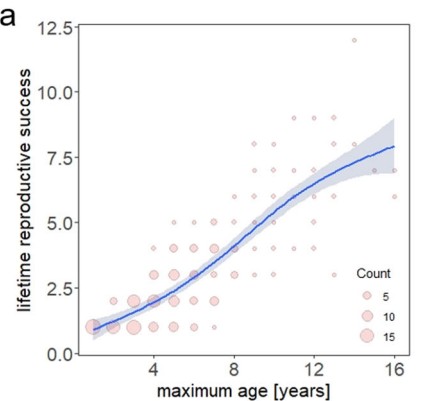
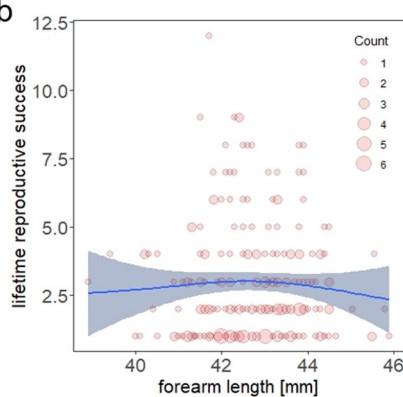

**Fig. 3 Influence of longevity and size on individual fitness.** Influence of (**a**) longevity and (**b**) forearm length [mm] on the lifetime reproductive success, as smooth functions of a GAM ($k = 6$, $n = 225$). Ninety-five per cent confidence intervals are added. Count gives the number of overlapping data points.

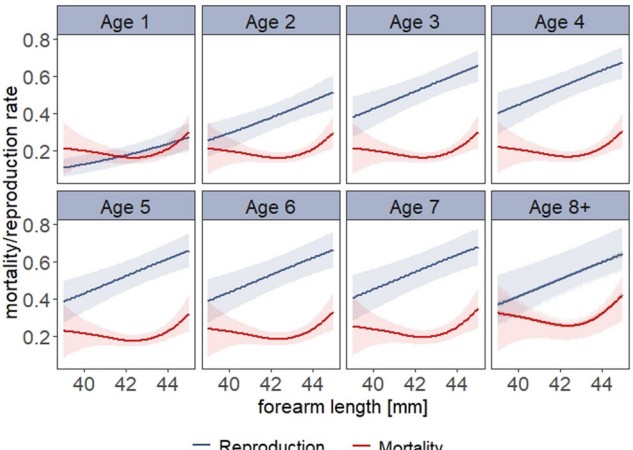

**Fig. 4 Trajectories of mortality and reproduction with body size.** Predicted response of the reproduction (blue) and mortality (red) increase in dependence of body size (forearm length (mm)). All underlying GAMs are calculated with $k = 10$, Gamma = 1.4 and method = REML.

(LRS) to remain remarkably stable among different-sized bats (Fig. 6). Our results suggest that the endangered Bechstein's bat can cope with the current level of climate warming by adopting a faster life history at a time scale that surpasses adaptation through selection.

**Body size, mortality, reproduction and, ultimately, fitness.** Our previous studies[25,26] showed that body size has a profound impact on mortality in female Bechstein's bats. Here, by adding fecundity and age at first reproduction to the models, we found that these life-history parameters are the drivers behind the observed mortality increases in larger bats. The age at first reproduction was strongly influenced by body size, with larger females reproducing earlier, a pattern that is common in several other long-lived mammals[32,33], but not ubiquitous. For example, in female greater horseshoe bats (*Rhinolophus ferrumequinum*), Ransome (1995)[30] reported no size difference in early and late breeders. In contrast, we found that larger bats were more likely to reproduce in a given year throughout their life. Larger females may be better able to maintain homeothermy during gestation[34], and larger mothers may also have a better energy balance when carrying their young or producing milk[35]. Although bats are generally described as 'income breeders', there is some evidence that larger individuals are more likely to arouse from hibernation

with 'leftover' fat reserves that they then can invest in reproduction[36,37]. Consequently, larger females that retain more leftover capital after emergence from hibernation might start reproduction at an earlier age and reproduce more likely in a given year. Our results suggest that body size impacts mortality by shaping age at first reproduction and fecundity, that are part of the longevity-reproduction-trade off.

We further found that females bred at younger ages in larger colonies. Besides summer temperature, colony size is an important determinant of body size, with juveniles growing larger in large colonies[25]. Given the high philopatry of female bats, an earlier onset of first reproduction in large colonies might be a consequence of the fact that bats are larger in large colonies, and that larger bats start reproduction earlier in life. However, as we included both, the body size as well as the colony size in the models, and colony size remained a significant factor, there must be additional benefits of living in larger colonies. Previous studies found that female Bechstein's bats and their offspring energetically benefit from social thermoregulation[38,39], which has also been shown for other bat species[40]. These thermoregulatory savings might translate into earlier reproduction.

We also found costs associated with reproduction in the form of reduced life expectancy in early reproducing bats. For comparison, Ransome (1995)[30] also reported reduced life-spans in early breeding female greater horseshoe bats. A review by Lemaître et al. (2015)[41] emphasizes the occurrence of early-late life trade-offs across a wide range of animal species, arguing that the allocation of limited resources to reproduction early in life leads to a loss of somatic maintenance in later life. Along this lines, Wilkinson & South (2002)[42] proposed that the observed association between life span and reproductive rate across bat species suggests that bat longevity is influenced by an allocation of 'non-renewable resources' to reproduction. Another widely reported cost of early reproduction is the trade-off with future growth[32,43,44]. However, this specific cost does not apply, as Bechstein's bats finish skeletal growth before they first reproduce (compare Culina et al., 2019)[45].

A potential cost of delaying reproducing to older ages is the so called Lansing effect, which states that parental age can affect the lifespan of offspring[46]. In many species, offspring born to older parents suffer increased mortality rates compared to offspring from younger parents[47]. While it remains unclear if the Lansing effect exists in bats, it could potentially favor an early onset of reproduction in female Bechstein's bats, which might further offset the disadvantage of increased mortality in larger individuals.

The maximisation of fitness lies at the heart of all life-history trade-offs and our data show that in female Bechstein's bats,

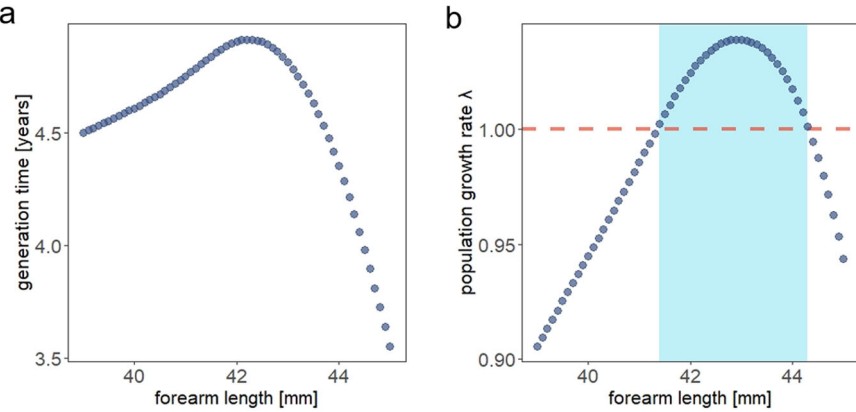

**Fig. 5 The effects of body size on generation times and population growth rate.** Predicted response of (**a**) generation time and (**b**) population growth rate in dependence of body size (forearm length (mm)). The highlighted blue area marks the size range, where population growth is predicted to occur. All underlying GAMs are calculated with $k = 10$, Gamma = 1.4 and method = REML.

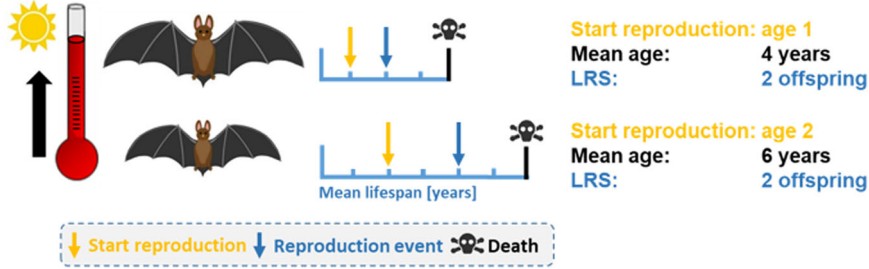

**Fig. 6 Graphical summary of the interplay between global warming, life-history traits and lifetime reproductive success.** With higher temperatures bats grow larger and compensate for their higher mortality with a faster pace of life.

lifetime reproductive success (LRS) is predominantly driven by longevity. This pattern is ubiquitous in long-lived mammals (e.g., 81% of variance explained by longevity in Thornicroft's giraffes (*Giraffa camelopardalis thornicrofti*)[28]; 70% in yellow baboons (*Papio cynoscephalus*))[29]. In the greater horseshoe bat, a staggering 96-99% of LRS was explained by maximum age alone[30]. Given this clear dependence of fitness on longevity, and given the observed increases of mortality in larger female Bechstein's bats[25,26], we had expected to find a reduced fitness in larger individuals. However, despite their reduced longevity, our results reveal that the LRS of larger bats did not differ from smaller bats. Consequently, the acceleration of their pace of life meant that larger females were able to compensate for their reduced longevity. Despite this optimistic finding, there was a trend towards reduced fitness in larger bats (see Fig. 3b). That, while presumably not problematic at the current time, could represent a problem for future populations if they consist of a higher proportion of larger bats as a result of increasing summer temperatures.

**Changes in life history**. Across species, age-specific mortality hazards and the spread of reproduction generally shape animal life history strategies[17]. Our study reveals that a long-lived mammal also displays plasticity in the pace of their life history, which may enable bats to adapt to changing temperatures. This plasticity can be attributed to the immediate link between individual body size and life history[25]. The life history of large females is characterized by faster generation times, an earlier age at first reproduction and a shorter lifetime expectancy. Predicting the developments under a global warming scenario, Bechstein's bat populations will consist of an increasing proportion of large

animals with faster life histories. Environmentally mediated changes in life-histories of species have been observed before. This phenomenon is often seen in fish populations, where strong fishing pressure shifts populations towards younger, smaller, and more quickly maturing individuals[48–50]. In Soay sheep (*Ovies aries*), Forchhammer et al. (2001)[22] reported inter-cohort variation in life-history traits after changes in the North Atlantic Oscillation Index, which integrates temperature and precipitation. However, change of the pace of life as a consequence of climate change (especially increased temperature) has only rarely been reported, predominantly in poikilothermic animals (in red spider mites[51]; fish[52], crickets[53]), but not in mammalian species on the slow-end of the fast-slow-continuum. Our results show the potential for a shift towards a faster life history, which in itself gives hope to faster response capacities to changing environments. Faster generation times might further support population growth for this slow-reproducing, endangered bats species. This is corroborated by our finding that population growth rates are currently highest in medium to large-sized bats, whereas at the extreme ends of the spectrum growth rates are predicted to revert into a decline.

Incorporating the summer temperature directly into the models, we were not able to detect a significant link between temperature and the examined traits. While temperature explains more than a third of the deviance of the body size (37%), there are clearly additional factors affecting growth in bats. This is also evident in the high residual variance of body sizes of bats in the size-temperature relationship (see Fig. 1). Some of those factors, as the colony size, have already been identified[25], while others, such as the heritability of body size, are currently under investigation or remain unknown. Considering these complex interactions, it is perhaps not surprising that we failed to find a direct link between demographic parameters

and temperature. Nonetheless, our data clearly show the link between higher summer temperatures and larger body sizes, and the subsequent impact of body size on the speed of life history paces. Thus, global warming indirectly induces a faster pace of life, by leading to larger bats.

Before moving on to predict population persistence under varying global warming scenarios, additional aspects must be considered: First, if large females reproduce more frequently and have fewer years between reproductive events, Bechstein's bat colonies will require a habitat quality that allows for annual reproduction. Secondly, small but longer-lived individuals may be better able to skip reproduction in years with bad weather conditions, with only limited cost. In contrast, larger and shorter-lived individuals would have reduced lifetime reproductive success, if they are not able to successfully raise an offspring because of poor environmental conditions. As environmental stochasticity in form of extreme weather events is expected to increase under climate change[27], a predominance of relative fast life histories in the Bechstein's bat populations might be detrimental in the long term.

## Methods

**Study site and data collection.** Morphological and demographic data were collected between 1996 and 2020 in four Bechstein's bat colonies living in forests near Würzburg, Germany[54]. Colonies comprise 10-45 adult females and their offspring (males are solitary). All adult colony members were individually marked with RFID-tags (Trovan, Germany)[55]. During two capture events each year, forearm length (FAL) was measured to the nearest 0.1 mm using callipers[56] and wing tissue samples were collected for genetic analysis and assignment of maternity[54]. For statistical analysis we used the first spring measurement of FAL for an individual. We determined colony size and survival of individuals based on presence-absence data obtained from the annual recaptures and roost monitoring with RFID-readers carried out each year from mid-April to September. Due to the almost daily roost monitoring and high natal philopatry of females[54], the census data allow for reliable survival and reproduction estimates[25,26]. In total, 331 females were followed from birth to death, of which 225 individuals reproduced at least once in their lifetime. Including individuals that were still alive at the end of the study period, demographic data were available for a total of 381 females.

**Summer temperature.** Data on ambient temperatures were provided by the Bayerische Landesanstalt für Wald und Forstwirtschaft (LWF) from the meteorological station 'Waldklimastation' (WKS; ID WUE[57]) and the DWD Climate Data Center (meteorological station 'Würzburg'; ID 05705[58]) and aggregated as a 24 h mean. We used the mean minimum temperature of the sensitive time window (June 22nd to July 16th), during which body size is most susceptible to variation in minimum temperature[25].

**Genetic analysis and assignment of offspring.** After DNA extraction, samples were genotyped using 13 polymorphic microsatellite loci, 12 of which had been previously established[59,60]. The last marker (GD7VI) was developed following the procedure described in van Schaik et al. (2018)[60]. The sequence and primer information have been submitted to GenBank and are available under ascension no. OL961308. We used CERVUS (3.0.7)[61] to estimate null allele frequencies, observed heterozygosity, expected heterozygosity, polymorphic information content, average non-exclusion probability, and test for Hardy-Weinberg equilibrium for all markers. Observed heterozygosities for the 13 microsatellite markers ranged from 0.64 to 0.91 with a combined non-exclusion probability (for first parent) of 0.0011. A detailed description of all markers and multiplexes used for PCR-amplification is given in the Supplementary Information (see Supplementary Table S6). PCR protocols followed van Schaik et al. 2018[60]. PCR products were run on a 3130 Genetic Analyzer (Applied Biosystems). Microsatellites were scored using GENEMAPPER software version 5.0 (ThermoFisher). We first estimated the relatedness between all possible mothers of each year for each colony using the 'TrioML' function of the software package Coancestry (1.0.1.9)[62] and then assigned parentage using CERVUS. We assigned parentage to only those mother-offspring pairs whose LOD-scores were significant at the strict 95% confidence level, and/or which had 0 mismatches. Out of 1,101 juveniles registered in the years 1996-2020, 51.8% were female. Of those 570 female juveniles, 497 (87.2%) were successfully assigned to mothers. A discussion on the robustness of our methodological approach in comparison with other approaches (i.e., using genetic matches regardless of confidence levels, or assessing lactation status to determine successful reproduction), can be found in the supplemental information (see Supplementary Note 1).

**Life-history traits.** Lifetime reproductive success (LRS) was determined as the total number of offspring successfully weaned by a female during her lifetime. LRS

was calculated only for females who were born into the study and completed their life before the end of the study (September 2020). Mature females alive at the start of the study in 1996 were not considered for the analysis. **Age at first reproduction** was the age a female had her first offspring, with mother-offspring pairs matched by the aforementioned parentage analysis. For each adult female we calculated **fecundity** as the LRS divided by the number of breeding seasons experienced by the respective female[18]. To calculate **generation time** (T), which is the average age of the mothers for all offspring produced by a single cohort[63], we followed the approach by Gotelli (2008)[63] using equation (1)

$$T = \frac{R0}{\sum l(x) * m(x)} \tag{1}$$

where l(x) is the probability of survival from birth to age x, and m(x) is the birth rate of female offspring to a female of age x. The net reproductive rate R0 was calculated[64] as equation (2):

$$R0 = \sum l(x) * m(x) * x \tag{2}$$

and was then used to estimate the **population growth rate**, λ, as equation (3)[63]:

$$\lambda = \exp\left(\frac{\ln(R0)}{T}\right) \tag{3}$$

λ is used to distinguish population decline (λ < 1) from population growth (λ > 1). To compare generation time and population growth rate across different body sizes, we used estimates obtained from GAMs (see below '**Reproductive rate**') for reproduction and mortality in specific age and size classes.

**Structure of datasets.** To investigate re-occurring annual events, such as the **reproductive rate** in a given year, we used a dataset that included year-by-year information on survival and reproduction over the lifespan of each individual. Individual ID entered as a random effect to account for individual heterogeneity[65] and pseudo-replication.

For lifetime-based analyses on summary measures, such as **lifetime reproductive success**, we used a reduced, comprehensive dataset where each individual entered the dataset only once.

**Model building.** All analyses were performed in R (Version 4.1.1)[66]. Using the 'gam' function in the 'mgcv' package[67] we built generalized additive models (GAMs) to study nonlinear impacts of age and size on different life-history traits. Models were calculated using either a 'poisson' or 'binomial' link function fitted with the restricted maximum likelihood (REML) method and a gamma value of 1.4 (see Supplementary Table S7 for an overview). Data distributions were verified using the 'fitdistrplus'-package[68]. We centred (by subtracting the mean) 'size' and 'maximum age' to reduce multicollinearity and used the Akaike Information Criterion (AIC[69]) for model selection. 'Body size' and 'age' entered as smooth functions to allow for non-linear fits. 'Colony ID' and either 'year' or 'birth year' entered as random factors, to control for possible differences among the four colony sites and the years of observation (see Supplementary Table S7). 'Age at first reproduction' was treated as a parametric variable. As age at first reproduction was found to be density-dependent in other studies[70] we included colony size as a measure of local population density. We defined 'colony size' as the number of females that were logged and/or caught in a given year in the roosts of the respective colony.

**Reproductive rate.** First, to gain a more detailed insight on what parameters shaped the observed rate of reproduction (model no. 4, see Supplementary Table S7), and to estimate effect sizes of those parameters, we built explanatory models that tested for the influence of colony size, age and body size.

Second, we modelled responses of reproductive rate and mortality across all different body sizes and age classes. To do so, we built predictions for the reproduction and mortality rates based on fitted GAMs using a 'response' type in the 'predict.gam' function of the package. These underlying GAMs had the same random structure as model 4 and 5 (Supplementary Table S7), using 'colony ID', 'year' and 'individual ID' as random effects, and were built on uncensored data that included all individuals regardless of their reproduction status or if they were still alive at the time of the study. Estimates of these predictive models for reproduction and mortality were used to then calculate responses of generation time (T) and population growth (λ) to body size (see '**Life history traits**').

**Cost of reproduction.** To analyze trade-offs between reproduction and mortality (model no. 5, see Supplementary Table S7), we built a discrete-time survival analysis framework similar to that of Fleischer et al. (2017)[26], but added the reproductive parameters 'age at first reproduction' as well as 'fecundity' as further explanatory variables. This required individuals that had complete life-histories (so we could estimate fecundity as the realized lifetime reproduction rate), as well as individuals that had reproduced at least once in their lifetime (so we could include 'age at first reproduction'). The winter of 2010 had an extremely high mortality rate due to an exceptional catastrophic weather event[26]. We thus repeated the mortality analysis with a dataset excluding the year 2010, to verify that the single catastrophic event was not driving the observed results. We found no differences compared to the full data set (see Supplementary Table S3a and b).

**Statistics and reproducibility**. All analyses were performed in R (Version 4.1.1)[66]. Statistical analyses were conducted using the above cited packages and reproducibility can be achieved using the functions and parameters described in the Methods. Additionally, an overview of the different models tested with details on the e.g. response variables and model structure is listed in the Supplementary Table S7. The code is available at Zenodo under https://doi.org/10.5281/zenodo.5878883.

**Ethics**. The handling, tagging and monitoring of the bats were conducted under permits for species protection (55.1.−8642.01-2/00) and animal welfare (55.2-DMS 2532-2-20) that had been issued by the government of Lower Franconia.

**Reporting summary**. Further information on research design is available in the Nature Research Reporting Summary linked to this article.

## Data availability

The datasets analysed during the current study are available from the corresponding author on reasonable request. Source data for graphs and charts are available under https://doi.org/10.5281/zenodo.6543599[71].

## Code availability

The code is available at Zenodo under https://doi.org/10.5281/zenodo.6543599[71].

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

## Acknowledgements
We would like to thank Jan Gogarten for valuable comments and proofreading of the manuscript. We also thank Colin O'Donnell and one anonymous reviewer for their helpful comments on a previous version of the manuscript. Finally, we thank numerous helpers in the field.

## Author contributions
G.K. conceived the project and acquired funding; CM and AS carried out the statistical analysis. CM and TF prepared the data. CM, GK and AS wrote the manuscript with support from TF. All authors gave final approval for publication and agreed to be held accountable for the work performed therein.

## Funding

## Competing interests
The authors declare no competing interests.
