## [Peer Review File · Communications Biology]

Reviewers' comments:

Reviewer #1 (Remarks to the Author):

I found this paper fascinating and it is rare that long term studies are available to demonstrate these influences on life-time reproductive success. The results show clearly that age of first breeding, fecundity and, to a lesser degree, colony size in Bechstein's bats are predicted by body size. The larger the bat, the earlier it breeds – but early breeders don't live as long as late breeders. The paper was well written and the statistical analyses presented seem sound.

Consequences for life-time reproductive success are clearly demonstrated and interestingly, these bats show faster and slower modes of reproduction, which the authors suggest provides a mechanism to adapt to climate change.

However, I thought the Title and Introduction of the paper promised analyses of a direct link between the authors' results and climate change ("Our study demonstrates that climate change can trigger a rapid shift to a faster pace of life, even in a mammal species with a slow life history"). The Introduction states "we assess how temperature induced changes in size affect demographic rates in wild Bechstein's bats" (Lines 29-30). While they speculate about this relationship in the Discussion, and they are likely correct in their assertions, there are no data that directly link the findings to climate change presented in this paper (e.g none of the models in the Supplementary data presented include temperature variables or time-temperature interactions).

Thus, assertions that "Global warming causes Bechstein's bats to produce larger females that face a higher mortality risk" and "global warming leads to a faster life history pace" are not supported directly by the analyses and should be viewed as worthwhile hypotheses for future investigation. Indeed, I don't see evidence in this paper that supports the authors assertion that "Our results show a shift towards a faster life history under increasing temperatures, which in itself gives hope to faster response capacities to changing environments" (Lines 302-304).

- Is there a correlation between increased temperatures and increased body size in these bats?
- Are bats breeding earlier, and is this correlated with body size?
- Is 'the pace of life' actually increasing in a systematic/predictive way with increased temperatures?
- Is mortality rate increasing in a similar (linear?) predictive way?

The author's do refer to an earlier paper (Mundinger et al 2021), which I don't have access to, that may contain analyses of some of these questions. If this is the case, then I think the links between the two papers, and how they might complement each other, needs to be clearer.

Data presented in this paper are definitely worth publishing, and the primary findings, that "Our data show that size-dependent fecundity and age at first reproduction drive the observed increase in mortality. Moreover, we found that the pace of life varies among females, with larger females having an earlier onset of reproduction, and shorter generation times" are sound and important.

The main finding in relation to climate change, is accurately reflected in the Discussion: "Our long-term field study suggests that a long-lived mammal species may be able to cope with global warming by speeding up their pace of life." However, analysing questions like those I pose above is required to justify the Title and make the link with climate change more direct – otherwise, the emphasis is more on implications of climate change and Discussion should change slightly to emphasise the demographic results and implications at the beginning. In the Conclusions, it is unclear which analyses provide justification of the Conclusions "Our study reveals a rapid switch to a faster pace of life in a slow-reproducing, long-lived mammal species" (Lines 322-323) and "Bechstein's bats offspring are larger in warmer summers" (Line 330).

Minor comments

Line 169 – is this a grand mean of forearm size +/- SE? – as forearm sizes would have been measured multiple times throughout the study.

Line 171 – How do you know these bats that 'died' simply didn't emigrate, as they were young nulliparous bats? If you can discount emigration, need to say so – otherwise the estimate of never breeding may be an over estimate.

Line 171 – presumably this statistic is +/- 1 SD?

Line 171 – "...mean forearm size..."

Table 1 – presume No. refers to 'model number' – better to be explicit

Table 1 – what is 'mom'??

Figure 1 – A little more detail in caption needed – what are the units for fecundity and age of first breeding? Was is "Count"? (so the Fig stands alone) Why wasn't Extended Data Figure 1 included with the 3 other graphs – would be good to have in one place.

Supplementary data – I would prefer Tables were listed in order that they are mentioned in text – e.g. First mention is Table S1 (line 70) but next mention of Supplementary data jumps to Table S5a)(line 163). Table S2 not mentioned until Line 207, but S3 -S6 mentioned before hand)

Reference

Mundinger, C., Scheuerlein, A. & Kerth, G. Long-term study shows that increasing body size in response to warmer summers is associated with higher mortality risk in a long-lived bat species. *Proc. R. Soc. B Biol. Sci.* 288, 417 20210508 (2021)

Colin O'Donnell

Reviewer #3 (Remarks to the Author):

This is an important and novel paper that highlights the value of long-term datasets for understanding how animal populations respond to climate change. The paper is therefore potentially of wide interest. The study is based on a 25-year study of Bechstein's bats, and has involved individual marking, reconstructions of pedigrees genetically, and collection of data on fitness parameters such as reproductive success and survival. The authors had previously shown how climate warming has led to large female bats being favoured, although these have a high mortality risk. Here they show how this risk is offset by the females breeding at a younger age, and hence in the long-term their lifetime reproductive success is not compromised. The paper shows how understanding life histories can be valuable for predicting how species of conservation concern may (or may not) be affected by climate change, and how species with slow histories (in this case one offspring at most is produced per year) can respond to climate change by adjusting more plastic traits (e.g. timing of first reproduction) rather than presumably less flexible traits such as litter size. The conclusions follow from the results, and I found the paper convincing.

I am curious about the roles of plasticity vs. selection here. It would be valuable to know whether traits such as timing of first breeding are heritable, and can thus respond to selection in the longer term. This sort of analysis may be possible given the pedigree data available. Plasticity from a repeated-measures perspective on the same individuals would be useful to understand.

Are there inclusive fitness costs associated with breeding earlier in life? For example, pups produced may be smaller, or born later, and ultimately have lower survival prospects. Therefore, there may be fitness consequences that extend beyond LRS.

What is role of timing of breeding within years? Is there an optimal time of year to give birth in terms of survival for both pups and mothers, and if so have larger females been breeding at more favourable times of year? Are data on timing of breeding within years available?

Why do females breed at younger ages in larger colony sizes? Could this be associated with thermoregulatory benefits?

Fig. 2a is interesting in that the highest values of LRS occur at intermediate body sizes – is this indicative of stabilising selection?

If mortality risk increases with age, does this suggest senescence is occurring?

Minor grammatical issues (generally the paper is well written).

Line 21. Rephrase to something like 'Bechstein's bats can reach 21 years of age, and during higher summer temperatures accelerate growth in juveniles, resulting in larger adult body sizes'.

Line 32 – be more specific, rather than using 'several hundred'.

Poisson should have capital P in table 1.

Reviewer #1 (Remarks to the Author):

I found this paper fascinating and it is rare that long term studies are available to demonstrate these influences on life-time reproductive success. The results show clearly that age of first breeding, fecundity and, to a lesser degree, colony size in Bechstein's bats are predicted by body size. The larger the bat, the earlier it breeds – but early breeders don't live as long as late breeders. The paper was well written and the statistical analyses presented seem sound.

Consequences for life-time reproductive success are clearly demonstrated and interestingly, these bats show faster and slower modes of reproduction, which the authors suggest provides a mechanism to adapt to climate change.

However, I thought the Title and Introduction of the paper promised analyses of a direct link between the authors' results and climate change ("Our study demonstrates that climate change can trigger a rapid shift to a faster pace of life, even in a mammal species with a slow life history"). The Introduction states "we assess how temperature induced changes in size affect demographic rates in wild Bechstein's bats" (Lines 29-30)". While they speculate about this relationship in the Discussion, and they are likely correct in their assertions, there are no data that directly link the findings to climate change presented in this paper (e.g none of the models in the Supplementary data presented include temperature variables or time-temperature interactions).

Thus, assertions that "Global warming causes Bechstein's bats to produce larger females that face a higher mortality risk" and "global warming leads to a faster life history pace" are not supported directly by the analyses and should be viewed as worthwhile hypotheses for future investigation. Indeed, I don't see evidence in this paper that supports the authors assertion that "Our results show a shift towards a faster life history under increasing temperatures, which in itself gives hope to faster response capacities to changing environments" (Lines 302-304).

We agree that this important point was not adequately addressed in our previous manuscript. To illustrate the role of temperature in this study, we now added a further plot in the Results (new Figure 1, lines 66-70), which clearly shows the influence of summer temperature on body size of offspring. We added to the manuscript in lines 66-70:

"Adult body size is known to be affected by the summer minimum temperature during a critical growth period in June-July (see Mundinger et al. 2021). Bats grow to larger sizes in warmer summers. This relationship is robust, and can also be found in a subset of the data used in this study, where only females with complete life histories are included (see Figure 1)".

And added the new Figure 1:

Figure 1: The influence of the summer minimum temperature [°C] during a critical time window in the birth year on the body size of bats (all bats with complete life histories, n=331). Count indicates the number of overlapping data points. The line and confidence interval indicate the smooth from a GAM of forearm length on summer temperature with colony ID and year as random factors.

In addition, we built and compared models that included summer temperature instead of body size, but failed to detect a direct influence of temperature on life history traits. Although temperature explains more than a third of the deviance of the body size (37%; new Figure 1), there are additional factors that influence growth in bats. This is also evident looking at the high variance of body size grown at the same summer temperature (see new Fig. 1). Some of the factors responsible for body size in Bechstein's bats, like the colony size, have already been identified, while others, such as the heritability of body size, are currently under investigation or remain unknown. And although summer temperatures have increased significantly over the study period, there is huge variance between years, with cold years interspersed between warmer ones (see Mundinger et al. 2021). Considering these complex interactions of factors, it is not surprising that we were not able to see a direct link between summer temperature and pace of life. Furthermore, especially in the higher temperature ranges (>14°C) we lack sufficient data to observe a direct connection. We have included this in the discussion (lines 227-239.). In summary, our data show that global warming acts indirectly on the pace of life by increasing the frequency of bats with large body sizes. Large bats in turn have a faster life history. To clarify this point, we changed the Title and Abstract.

- Is there a correlation between increased temperatures and increased body size in these bats?

Yes, there is a strong impact of summer temperatures on the body size of bats. This has been published as the main finding in Mundinger et al. (2021). We redid the

analysis for the subset used in this study and added a figure in the Results depicting the direct impact of temperature on the body size in the data set relevant for this study (new Figure 1, lines 66-70).

- Are bats breeding earlier, and is this correlated with body size?

The timing of breeding in *Myotis bechsteinii* is strongly influenced by spring temperatures, a pattern we also reported in a past publication (see Munding et al. 2021). However, we did not detect a trend towards earlier breeding dates throughout the entire 25-year study period. At the same time, we have not yet analyzed trends in spring temperatures over the last decades, and do not know yet of a directional change in spring temperatures. While this is beyond the scope of the current study, we will investigate the timing of parturition in a future paper on the phenology of Bechstein's bats.

- Is „the pace of life” actually increasing in a systematic/predictive way with increased temperatures?

We see a directional trend in the speed of the pace of life over the years with a concomitant increase in temperatures (see Figure 1 of our Proc R Soc B paper, Munding et al. (2021)), with increasing temperatures. Both the age at first reproduction (AFR) as well as the fecundity change over the study period, with animals reproducing at increasingly earlier ages, while the fecundity is increasing over the years.

- Is mortality rate increasing in a similar (linear?) predictive way?

The mortality rates for the study periods 1996-2014 have been published in Fleischer et al. (2017), and show no systematic increase. Also, with our updated dataset we do not find a directional change in mortality rates. However, a failure to detect a systematic increase of mortality over the years is not surprising, as the population each year consists of a mix of bats with varying body sizes. And as mentioned above, although summer temperatures have increased significantly over the study period, warm years are still regularly interspersed between cold years, which leads to a reset of body sizes. In addition, mortality in Bechstein's bats is predominantly determined by rare catastrophic events, rather than average temperature (Fleischer et al. 2017).

- The author's do refer to an earlier paper (Munding et al 2021), which I don't have access to, that may contain analyses of some of these questions. If this is the case, then I think the links between the two papers, and how they might complement each other, needs to be clearer.

We followed the advice and emphasized the link more strongly (see for example lines 61-70 as stated above). We additionally changed the Title (it now reads: "Global warming leads to larger bats with a faster life history pace in the long-lived Bechstein's bat (*Myotis bechsteinii*) and Abstract:

"Whether species can cope with environmental change depends considerably on their life history. Bats have long lifespans and low reproductive rates which make them vulnerable to

*environmental changes. Global warming causes Bechstein's bats (*Myotis bechsteinii*) to produce larger females that face a higher mortality risk. Here, we test whether these larger females are able to offset their elevated mortality risk by adopting a faster life history. We analysed an individual-based 25-year dataset from 331 RFID-tagged wild bats and combine genetic pedigrees with data on survival, reproduction and body size. We find that sizedependent fecundity and age at first reproduction drive the observed increase in mortality. Because larger females have an earlier onset of reproduction and shorter generation times, lifetime reproductive success remains remarkably stable among individuals with different body sizes. Our study demonstrates a rapid shift to a faster pace of life in a mammal with a slow life history."*

And added lines 227-239 in the Discussion:

"Incorporating the summer temperature directly into the models, we were not able to detect a significant link between temperature and the examined traits. While temperature explains more than a third of the deviance of the body size (37%), there are clearly additional factors affecting growth in bats. This is also evident in the high residual variance of body sizes of bats in the size – temperature relationship (see Figure 1). Some of those factors, as the colony size, have already been identified, while others, such as the heritability of body size, are currently under investigation or remain unknown. Considering these complex interactions, it is perhaps not surprising that we failed to find a direct link between demographic parameters and temperature. Nonetheless, our data clearly show the link between higher summer temperatures and larger body sizes, and the subsequent impact of body size on the speed of life history paces. Thus, global warming indirectly induces a faster pace of life, by leading to larger bats."

- Data presented in this paper are definitely worth publishing, and the primary findings, that "Our data show that size-dependent fecundity and age at first reproduction drive the observed increase in mortality. Moreover, we found that the pace of life varies among females, with larger females having an earlier onset of reproduction, and shorter generation times" are sound and important.

Thank you for the kind comments!

- The main finding in relation to climate change, is accurately reflected in the Discussion: "Our long-term field study suggests that a long-lived mammal species may be able to cope with global warming by speeding up their pace of life." However, analysing questions like those I pose above is required to justify the Title and make the link with climate change more direct – otherwise, the emphasis is more on implications of climate change and Discussion should change slightly to emphasise the demographic results and implications at the beginning. In the Conclusions, it is unclear which analyses provide justification of the Conclusions "Our study reveals a rapid switch to a faster pace of life in a slow-reproducing, long-lived mammal species" (Lines 322-323) and "Bechstein's bats offspring are larger in warmer summers" (Line 330).

As stated above we changed the Title, Abstract and Discussion to clarify on the link between climate change (temperature), body size and the effects of body size on the pace of life.

- Minor comments
- Line 169 – is this a grand mean of forearm size +/- SE? – as forearm sizes would have been measured multiple times throughout the study.

For the measurement `forearm size` we always used the first measurement of an individual when it was captured as an adult. Thus, only one measurement for each adult individual is used. We included this in line 260-261:

“For statistical analysis we used the first spring measurement of FAL for an individual.”

- Line 171 – How do you know these bats that „died“ simply didn’t emigrate, as they were young nulliparous bats? If you can discount emigration, need to say so – otherwise the estimate of never breeding may be an over estimate.

We elaborate on this in the Methods in the “Study site and data collection” section. Female Bechstein bats are highly philopatric, so we can indeed discount emigration.

- Line 171 – presumably this statistic is +/- 1 SD?

Yes, we added “SD” in the paragraph (lines 60-75):

“Average forearm length of all adult females with complete life histories (n=331) was 42.7 mm (+/- 1.2 mm SD).” [...] “Among reproducing females (n=225), the first reproduction event occurred at an average age of 2.2 years (+/- 1.1 years SD). Of all adult females, 32% never reproduced (those females died at an age of 1.7 years (+/- 1.1 years SD), with a mean forearm size of 42.8 mm).”

- Line 171 – “...mean forearm size...”

Done (line 74-75): *“Of all adult females, 32% never reproduced (those females died at an age of 1.7 years (+/- 1.1 years SD), with a mean forearm size of 42.8 mm).”*

- Table 1 – presume No. refers to „model number“ – better to be explicit

Done (Supplementary Information, Table S7)

- Table 1 – what is „mom“??

We changed it to “mother” (Supplementary Information, Table S7)

- Figure 1 – A little more detail in caption needed – what are the units for fecundity and age of first breeding? Was is “Count”? (so the Fig stands alone) Why wasn’t Extended Data Figure 1 included with the 3 other graphs – would be good to have in one place.

Supplementary data – I would prefer Tables were listed in order that they are mentioned in text – e.g. First mention is Table S1 (line 70) but next mention of Supplementary data jumps to Table S5a)(line 163). Table S2 not mentioned until Line 207, but S3 -S6 mentioned before hand)

Thank you for the advice. We have moved the previously Extended Data Figure 1 into the main Results (now included in Figure 2 as plot (b), lines 89-90) and changed the numeration of Tables to follow the order they are mentioned in the text. We further added more detail in the caption. Figure 2 and caption now read as following:

Fig.2: Impact of (**a**) body size and (**b**) colony size (as the number of adult females in a colony) on the age of first reproduction (as age in years); and the impact of (**c**) body size and (**d**) age at first reproduction on the fecundity (as the lifetime reproductive success divided by the number of breeding seasons) as smooth functions of a GAM ($k=6$, $n=225$). Ninety-five per cent confidence intervals are added. Count gives the number of overlapping data points.

Reference

Mundinger, C., Scheuerlein, A. & Kerth, G. Long-term study shows that increasing body size in response to warmer summers is associated with higher mortality risk in a long-lived bat species. *Proc. R. Soc. B Biol. Sci.* 288, 417 20210508 (2021)

Colin O'Donnell

Reviewer #3 (Remarks to the Author):

This is an important and novel paper that highlights the value of long-term datasets for understanding how animal populations respond to climate change. The paper is therefore potentially of wide interest. The study is based on a 25-year study of Bechstein's bats, and has involved individual marking, reconstructions of pedigrees genetically, and collection of data on fitness parameters such as reproductive success and survival. The authors had previously shown how climate warming has led to large female bats being favoured, although these have a high mortality risk. Here they show how this risk is offset by the females breeding at a younger

age, and hence in the long-term their lifetime reproductive success is not compromised. The paper shows how understanding life histories can be valuable for predicting how species of conservation concern may (or may not) be affected by climate change, and how species with slow histories (in this case one offspring at most is produced per year) can respond to climate change by adjusting more plastic traits (e.g. timing of first reproduction) rather than presumably less flexible traits such as litter size. The conclusions follow from the results, and I found the paper convincing.

Thank you for the kind comments!

- I am curious about the roles of plasticity vs. selection here. It would be valuable to know whether traits such as timing of first breeding are heritable, and can thus respond to selection in the longer term. This sort of analysis may be possible given the pedigree data available. Plasticity from a repeated-measures perspective on the same individuals would be useful to understand.

We certainly agree that estimating heritability of these traits would be highly interesting and definitely conservation relevant. Indeed, we are close to submitting a paper on the heritability of body size in Bechstein's bat and how heritability varies between different environments. The heritability of traits such as the timing of first breeding would be a very interesting follow-up study!

- Are there inclusive fitness costs associated with breeding earlier in life? For example, pups produced may be smaller, or born later, and ultimately have lower survival prospects. Therefore, there may be fitness consequences that extend beyond LRS.

Again, this is an interesting point and of course it is correct that there may be transgenerational effects. However, because of the longevity of our study species, sample size of females with a complete life history is dramatically reduced (and biased towards short-lived individuals) once we take the LRS of the daughters from females with a known birth-date and a completed LRS into account. We therefore hope to be able to address this important question in a future study.

- What is role of timing of breeding within years? Is there an optimal time of year to give birth in terms of survival for both pups and mothers, and if so have larger females been breeding at more favourable times of year? Are data on timing of breeding within years available?

The variance of the individual timing of breeding in a colony within a given year is very low, as parturition is highly synchronized between individuals. The high synchrony in breeding time is common in European bat species: "When bats give birth, they do so with exceptional synchronization. The spring temperatures and the first insects stimulate ovulation, and all females in the colony give birth to their young within the space of a few day" (Eklöf J., Rydell J. (2017) Reproduction. In: Bats. Springer, Cham. https://doi.org/10.1007/978-3-319-66538-2_6). Thus, larger females would at most show a difference in parturition dates of a few days, which we don't expect to make a difference regarding favourable/unfavourable conditions.

However, the variance among years, and to a lesser extent among colonies, is much more variable. As cited above, the timing of breeding in *Myotis bechsteinii* is highly

influenced by spring temperatures, a pattern we also reported in a previous publication (see Mundinger et al. 2021). We completely agree, that the timing of breeding is very likely to impact especially juvenile survival and that an optimal time window should exist, which we are currently investigating in another study.

- Why do females breed at younger ages in larger colony sizes? Could this be associated with thermoregulatory benefits?

Very good questions! Besides summer temperature, colony size is the second important factor shaping the body size of juveniles, with juveniles growing larger in large colonies. Consequently, bats breeding earlier in larger colonies could be simply a result of collinearity. However, as we included both, the body size as well as the colony size in the models, and colony size remains a significant factor, there must be additional benefits of living in larger colonies. Indeed, we also suspect that this has to do with the mentioned thermoregulatory benefits. In previous papers, we could show that female Bechstein's bats and their offspring benefit energetically from social thermoregulation (Pretzlaff et al. 2010; Küpper et al. 2016). This has also been shown for other bat species (Willis and Brigham 2007). We expanded the discussion (line 156-166) to incorporate this point:

"We further found that females bred at younger ages in larger colonies. Besides summer temperature, colony size is an important determinant of body size, with juveniles growing larger in large colonies. Given the high philopatry of female bats, an earlier onset of first reproduction in large colonies might be a consequence of the fact that bats are larger in large colonies, and that larger bats start reproduction earlier in life. However, as we included both, the body size as well as the colony size in the models, and colony size remained a significant factor, there must be additional benefits of living in larger colonies. Previous studies found that female Bechstein's bats and their offspring energetically benefit from social thermoregulation,, which has also been shown for other bat species. These thermoregulatory savings might translate into earlier reproduction."

- Fig. 2a is interesting in that the highest values of LRS occur at intermediate body sizes – is this indicative of stabilising selection?

We agree that this is a possible explanation of the described pattern of LRS if there are no intergenerational effects of sorts, as you mention above. However, as summer temperature is a driver of body size, the population is pushed towards the right side with lower LRS, so that stabilizing selection is not effective. Unfortunately, currently we cannot be sure about this, as the sample size is still not high enough to find a significant higher LRS for bats of intermediate body sizes. Thus, this remains speculative at present.

- If mortality risk increases with age, does this suggest senescence is occurring?

While in a previous study (Fleischer et al., 2017) no sign of senescence was detected in the Bechstein's bats, we found that with an updated, larger dataset that (old) age became a significant factor in determining mortality risk (see Mundinger et al. 2021).

This was likely a consequence of a higher number of „old“ individuals in the sample due to an increased length of the study.

- Minor grammatical issues (generally the paper is well written).
- Line 21. Rephrase to something like „Bechstein“s bats can reach 21 years of age, and during higher summer temperatures accelerate growth in juveniles, resulting in larger adult body sizes“.

We reworded the sentence following your suggestion (lines 22-24):

*“Bechstein’s bats (*Myotis bechsteinii*) can reach 21 years of age, and grow to larger adult body sizes in summers with higher temperatures during a sensitive period.”*

- Line 32 – be more specific, rather than using „several hundred“.

Done. We inserted the exact number, 331 individuals (line 33):

“We use a longitudinal, individual-based 25-year dataset of 331 RFID-tagged bats, combining genetic pedigrees with long-term data on survival, reproduction and body size.”

- Poisson should have capital P in table 1.

Done.

REVIEWERS' COMMENTS:

Reviewer #1 (Remarks to the Author):

Thanks for the opportunity to review this revised MS.

The authors have done a good job of addressing the issues I raised in my initial review. I am happy with the revised manuscript.

A couple of minor points:

1. Lines 45-60 read like Conclusions and Discussion rather than an Introduction and are repetitive of what is said later – move to a Conclusions section at end?
2. Now there is a new Figure 1 (Line 69), need to revise subsequent Figure numbers (Line 82 onwards).

Reviewer #3 (Remarks to the Author):

The suggestions I made have been implemented, or identified as topics under work in other manuscripts. I spotted a typo at line 297 ('Discussion').

Reviewer #1 (Remarks to the Author):

Thanks for the opportunity to review this revised MS.

The authors have done a good job of addressing the issues I raised in my initial review. I am happy with the revised manuscript.

A couple of minor points:

1. Lines 45-60 read like Conclusions and Discussion rather than an Introduction and are repetitive of what is said later – move to a Conclusions section at end?

The “Style and formatting guide” of the communications journals reads under the topic ‘Introduction (mandatory)’: “The final paragraph should be a brief summary of the major results and conclusions”. To follow this guide we had moved the mentioned lines in the first revision into the Introduction and would prefer to leave them there.

2. Now there is a new Figure 1 (Line 69), need to revise subsequent Figure numbers (Line 82 onwards).

We corrected the references to the Figure numbers (lines 78-82).

Reviewer #3 (Remarks to the Author):

The suggestions I made have been implemented, or identified as topics under work in other manuscripts. I spotted a typo at line 297 ('Discussion').

Typo corrected (line 125)